# Phased LSTM: Accelerating Recurrent Network Training for Long or Event-based Sequences

**Daniel Neil, Michael Pfeiffer, and Shih-Chii Liu**
Institute of Neuroinformatics
University of Zurich and ETH Zurich
Zurich, Switzerland 8057
{dneil, pfeiffer, shih}@ini.uzh.ch

## Abstract

Recurrent Neural Networks (RNNs) have become the state-of-the-art choice for extracting patterns from temporal sequences. However, current RNN models are ill-suited to process irregularly sampled data triggered by events generated in continuous time by sensors or other neurons. Such data can occur, for example, when the input comes from novel event-driven artificial sensors that generate sparse, asynchronous streams of events or from multiple conventional sensors with different update intervals. In this work, we introduce the Phased LSTM model, which extends the LSTM unit by adding a new time gate. This gate is controlled by a parametrized oscillation with a frequency range that produces updates of the memory cell only during a small percentage of the cycle. Even with the sparse updates imposed by the oscillation, the Phased LSTM network achieves faster convergence than regular LSTMs on tasks which require learning of long sequences. The model naturally integrates inputs from sensors of arbitrary sampling rates, thereby opening new areas of investigation for processing asynchronous sensory events that carry timing information. It also greatly improves the performance of LSTMs in standard RNN applications, and does so with an order-of-magnitude fewer computes at runtime.

## 1 Introduction

Interest in recurrent neural networks (RNNs) has greatly increased in recent years, since larger training databases, more powerful computing resources, and better training algorithms have enabled breakthroughs in both processing and modeling of temporal sequences. Applications include speech recognition [13], natural language processing [1, 20], and attention-based models for structured prediction [5, 29]. RNNs are attractive because they equip neural networks with memories, and the introduction of gating units such as LSTM and GRU [16, 6] has greatly helped in making the learning of these networks manageable. RNNs are typically modeled as discrete-time dynamical systems, thereby implicitly assuming a constant sampling rate of input signals, which also becomes the update frequency of recurrent and feed-forward units. Although early work such as [25, 10, 4] has realized the resulting limitations and suggested continuous-time dynamical systems approaches towards RNNs, the great majority of modern RNN implementations uses fixed time steps.

Although fixed time steps are perfectly suitable for many RNN applications, there are several important scenarios in which constant update rates impose constraints that affect the precision and efficiency of RNNs. Many real-world tasks for autonomous vehicles or robots need to integrate input from a variety of sensors, e.g. for vision, audition, distance measurements, or gyroscopes. Each sensor may have its own data sampling rate, and short time steps are necessary to deal with sensors with high sampling frequencies. However, this leads to an unnecessarily higher computational load and

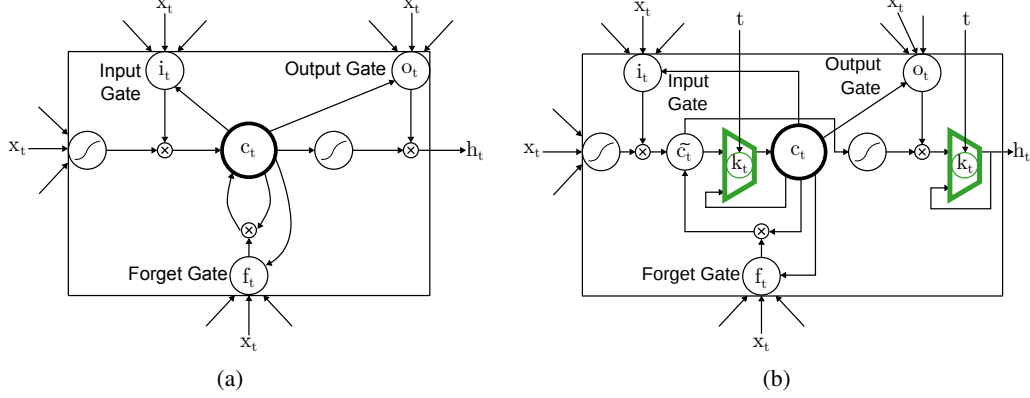

Figure 1: Model architecture. **(a)** Standard LSTM model. **(b)** Phased LSTM model, with time gate $k_t$ controlled by timestamp $t$. In the Phased LSTM formulation, the cell value $c_t$ and the hidden output $h_t$ can only be updated during an "open" phase; otherwise, the previous values are maintained.

power consumption so that all units in the network can be updated with one time step. An interesting new application area is processing of event-based sensors, which are data-driven, and record stimulus changes in the world with short latencies and accurate timing. Processing the asynchronous outputs of such sensors with time-stepped models would require high update frequencies, thereby counteracting the potential power savings of event-based sensors. And finally there is an interest coming from computational neuroscience, since brains can be viewed loosely as very large RNNs. However, biological neurons communicate with spikes, and therefore perform asynchronous, event-triggered updates in continuous time. This work presents a novel RNN model which can process inputs sampled at asynchronous times and is described further in the following sections.

## 2 Model Description

Long short-term memory (LSTM) units [16] (Fig. 1(a)) are an important ingredient for modern deep RNN architectures. We first define their update equations in the commonly-used version from [12]:

$$i_t = \sigma_i(x_t W_{xi} + h_{t-1} W_{hi} + w_{ci} \odot c_{t-1} + b_i) \tag{1}$$
$$f_t = \sigma_f(x_t W_{xf} + h_{t-1} W_{hf} + w_{cf} \odot c_{t-1} + b_f) \tag{2}$$
$$c_t = f_t \odot c_{t-1} + i_t \odot \sigma_c(x_t W_{xc} + h_{t-1} W_{hc} + b_c) \tag{3}$$
$$o_t = \sigma_o(x_t W_{xo} + h_{t-1} W_{ho} + w_{co} \odot c_t + b_o) \tag{4}$$
$$h_t = o_t \odot \sigma_h(c_t) \tag{5}$$

The main difference to classical RNNs is the use of the gating functions $i_t$, $f_t$, $o_t$, which represent the *input*, *forget*, and *output* gate at time $t$ respectively. $c_t$ is the cell activation vector, whereas $x_t$ and $h_t$ represent the input feature vector and the hidden output vector respectively. The gates use the typical sigmoidal nonlinearities $\sigma_i$, $\sigma_f$, $\sigma_o$ and tanh nonlinearities $\sigma_c$, and $\sigma_h$ with weight parameters $W_{hi}$, $W_{hf}$, $W_{ho}$, $W_{xi}$, $W_{xf}$, and $W_{xo}$, which connect the different inputs and gates with the memory cells and outputs, as well as biases $b_i$, $b_f$, and $b_o$. The cell state $c_t$ itself is updated with a fraction of the previous cell state that is controlled by $f_t$, and a new input state created from the element-wise (Hadamard) product, denoted by $\odot$, of $i_t$ and the output of the cell state nonlinearity $\sigma_c$. Optional *peephole* [11] connection weights $w_{ci}$, $w_{cf}$, $w_{co}$ further influence the operation of the input, forget, and output gates.

The Phased LSTM model extends the LSTM model by adding a new *time gate*, $k_t$ (Fig. 1(b)). The opening and closing of this gate is controlled by an independent rhythmic oscillation specified by three parameters; updates to the cell state $c_t$ and $h_t$ are permitted only when the gate is open. The first parameter, $\tau$, controls the real-time period of the oscillation. The second, $r_{on}$, controls the ratio of the duration of the "open" phase to the full period. The third, $s$, controls the phase shift of the oscillation to each Phased LSTM cell. All parameters can be learned during the training process. Though other variants are possible, we propose here a particularly successful linearized formulation

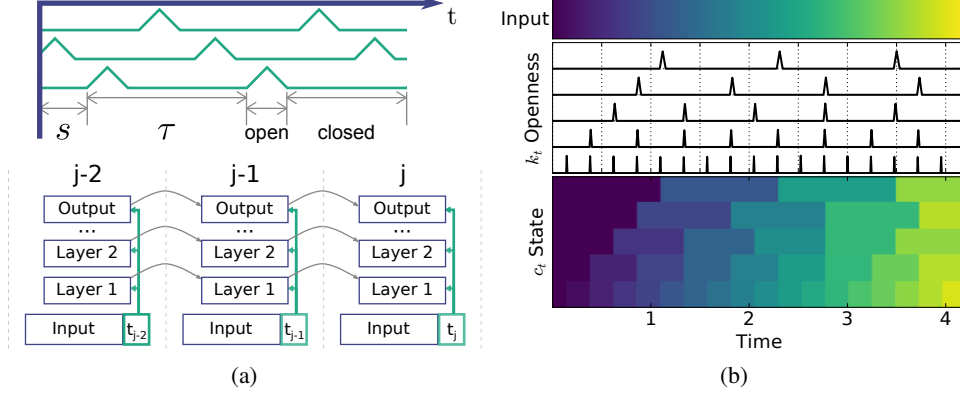

<center>(a)</center>

<center>(b)</center>

Figure 2: Diagram of Phased LSTM behaviour. **(a)** Top: The rhythmic oscillations to the time gates of 3 different neurons; the period $\tau$ and the phase shift $s$ is shown for the lowest neuron. The parameter $r_{on}$ is the ratio of the open period to the total period $\tau$. Bottom: Note that in a multilayer scenario, the timestamp is distributed to all layers which are updated at the same time point. **(b)** Illustration of Phased LSTM operation. A simple linearly increasing function is used as an input. The time gate $k_t$ of each neuron has a different $\tau$, identical phase shift $s$, and an open ratio $r_{on}$ of 0.05. Note that the input (top panel) flows through the time gate $k_t$ (middle panel) to be held as the new cell state $c_t$ (bottom panel) only when $k_t$ is open.

of the time gate, with analogy to the rectified linear unit that propagates gradients well:

$$\phi_t = \frac{(t - s) \bmod \tau}{\tau}, \qquad k_t = \begin{cases} \dfrac{2\phi_t}{r_{on}}, & \text{if } \phi_t < \dfrac{1}{2}r_{on} \\[2ex] 2 - \dfrac{2\phi_t}{r_{on}}, & \text{if } \dfrac{1}{2}r_{on} < \phi_t < r_{on} \\[2ex] \alpha\phi_t, & \text{otherwise} \end{cases} \tag{6}$$

$\phi_t$ is an auxiliary variable, which represents the phase inside the rhythmic cycle. The gate $k_t$ has three phases (see Fig. 2a): in the first two phases, the "openness" of the gate rises from 0 to 1 (first phase) and drops from 1 to 0 (second phase). During the third phase, the gate is closed and the previous cell state is maintained. The leak with rate $\alpha$ is active in the closed phase, and plays a similar role as the leak in a parametric "leaky" rectified linear unit [15] by propagating important gradient information even when the gate is closed. Note that the linear slopes of $k_t$ during the open phases of the time gate allow effective transmission of error gradients.

In contrast to traditional RNNs, and even sparser variants of RNNs [19], updates in Phased LSTM can optionally be performed at irregularly sampled time points $t_j$. This allows the RNNs to work with event-driven, asynchronously sampled input data. We use the shorthand notation $c_j = c_{t_j}$ for cell states at time $t_j$ (analogously for other gates and units), and let $c_{j-1}$ denote the state at the previous update time $t_{j-1}$. We can then rewrite the regular LSTM cell update equations for $c_j$ and $h_j$ (from Eq. 3 and Eq. 5), using proposed cell updates $\widetilde{c_j}$ and $\widetilde{h_j}$ mediated by the time gate $k_j$:

$$\widetilde{c_j} = f_j \odot c_{j-1} + i_j \odot \sigma_c(x_j W_{xc} + h_{j-1} W_{hc} + b_c) \tag{7}$$

$$c_j = k_j \odot \widetilde{c_j} + (1 - k_j) \odot c_{j-1} \tag{8}$$

$$\widetilde{h_j} = o_j \odot \sigma_h(\widetilde{c_j}) \tag{9}$$

$$h_j = k_j \odot \widetilde{h_j} + (1 - k_j) \odot h_{j-1} \tag{10}$$

A schematic of Phased LSTM with its parameters can be found in Fig. 2a, accompanied by an illustration of the relationship between the time, the input, the time gate $k_t$, and the state $c_t$ in Fig. 2b.

One key advantage of this Phased LSTM formulation lies in the rate of memory decay. For the simple task of keeping an initial memory state $c_0$ as long as possible without receiving additional inputs (i.e. $i_j = 0$ at all time steps $t_j$), a standard LSTM with a nearly fully-opened forget gate (i.e. $f_j = 1 - \epsilon$) after $n$ update steps would contain

$$c_n = f_n \odot c_{n-1} = (1 - \epsilon) \odot (f_{n-1} \odot c_{n-2}) = \ldots = (1 - \epsilon)^n \odot c_0 \quad . \tag{11}$$

<center>3</center>

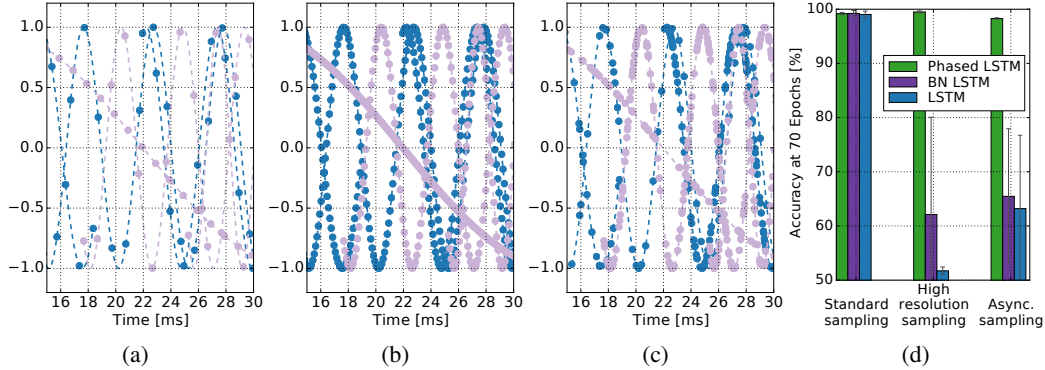

Figure 3: Frequency discrimination task. The network is trained to discriminate waves of different frequency sets (shown in blue and gray); every circle is an input point. **(a)** Standard condition: the data is regularly sampled every 1 ms. **(b)** High resolution sampling condition: new input points are gathered every 0.1ms. **(c)** Asynchronous sampling condition: new input points are presented at intervals of 0.02 ms to 10 ms. **(d)** The accuracy of Phased LSTM under the three sampling conditions is maintained, but the accuracy of the BN-LSTM and standard LSTM drops significantly in the sampling conditions (b) and (c). Error bars indicate standard deviation over 5 runs.

This means the memory for $\epsilon < 1$ decays exponentially with every time step. Conversely, the Phased LSTM state only decays during the open periods of the time gate, but maintains a perfect memory during its closed phase, i.e. $c_j = c_{j-\Delta}$ if $k_t = 0$ for $t_{j-\Delta} \leq t \leq t_j$. Thus, during a single oscillation period of length $\tau$, the units only update during a duration of $r_{on} \cdot \tau$, which will result in substantially fewer than $n$ update steps. Because of this cyclic memory, Phased LSTM can have much longer and adjustable memory length via the parameter $\tau$.

The oscillations impose sparse updates of the units, therefore substantially decreasing the total number of updates during network operation. During training, this sparseness ensures that the gradient is required to backpropagate through fewer updating timesteps, allowing an undecayed gradient to be backpropagated through time and allowing faster learning convergence. Similar to the shielding of the cell state $c_t$ (and its gradient) by the input gates and forget gates of the LSTM, the time gate prevents external inputs and time steps from dispersing and mixing the gradient of the cell state.

## 3 Results

In the following sections, we investigate the advantages of the Phased LSTM model in a variety of scenarios that require either precise timing of updates or learning from a long sequence. For all the results presented here, the networks were trained with Adam [18] set to default learning rate parameters, using Theano [2] with Lasagne [9]. Unless otherwise specified, the leak rate was set to $\alpha = 0.001$ during training and $\alpha = 0$ during test. The phase shift, $s$, for each neuron was uniformly chosen from the interval $[0, \tau]$. The parameters $\tau$ and $s$ were learned during training, while the open ratio $r_{on}$ was fixed at $0.05$ and not adjusted during training, except in the first task to demonstrate that the model can train successfully while learning all parameters.

### 3.1 Frequency Discrimination Task

In this first experiment, the network is trained to distinguish two classes of sine waves from different frequency sets: those with a period in a target range $T \sim \mathcal{U}(5,6)$, and those outside the range, i.e. $T \sim \{\mathcal{U}(1,5) \cup \mathcal{U}(6,100)\}$, using $\mathcal{U}(a,b)$ for the uniform distribution on the interval $(a,b)$. This task illustrates the advantages of Phased LSTM, since it involves a periodic stimulus and requires fine timing discrimination. The inputs are presented as pairs $\langle y,\ t \rangle$, where $y$ is the amplitude and $t$ the timestamp of the sample from the input sine wave.

Figure 3 illustrates the task: the blue curves must be separated from the lighter curves based on the samples shown as circles. We evaluate three conditions for sampling the input signals: In the standard condition (Fig. 3a), the sine waves are regularly sampled every 1 ms; in the oversampled

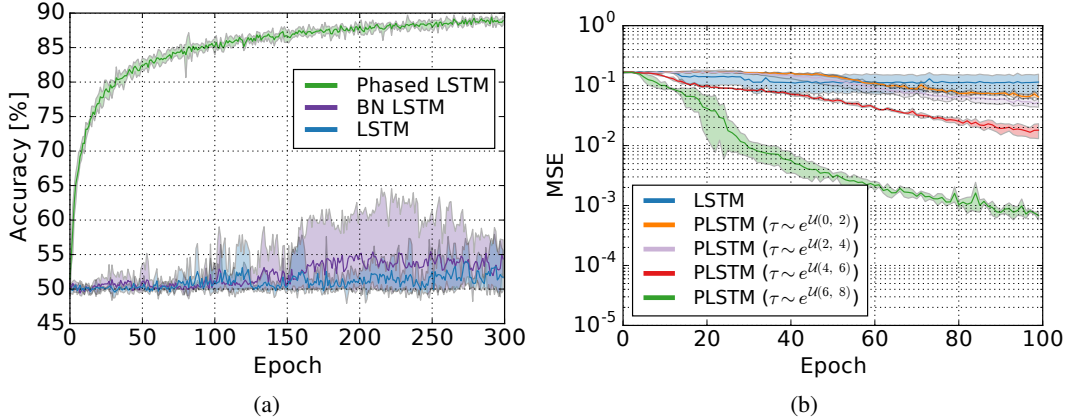

Figure 4: **(a)** Accuracy during training for the superimposed frequencies task. The Phased LSTM outperforms both LSTM and BN-LSTM while exhibiting lower variance. Shading shows maximum and minimum over 5 runs, while dark lines indicate the mean. **(b)** Mean-squared error over training on the addition task, with an input length of 500. Note that longer periods accelerate learning convergence.

condition (Fig. 3b), the sine waves are regularly sampled every 0.1 ms, resulting in ten times as many data points. Finally, in the asynchronously sampled condition (Fig. 3c), samples are collected at asynchronous times over the duration of the input. Additionally, the sine waves have a uniformly drawn random phase shift from all possible shifts, random numbers of samples drawn from $\mathcal{U}(15, 125)$, a random duration drawn from $\mathcal{U}(15, 125)$, and a start time drawn from $\mathcal{U}(0, 125 - $ duration). The number of samples in the asynchronous and standard sampling condition is equal. The classes were approximately balanced, yielding a 50% chance success rate.

Single-layer RNNs are trained on this data, each repeated with five random initial seeds. We compare our Phased LSTM configuration to regular LSTM, and batch-normalized (BN) LSTM which has found success in certain applications [14]. For the regular LSTM and the BN-LSTM, the timestamp is used as an additional input feature dimension; for the Phased LSTM, the time input controls the time gates $k_t$. The architecture consists of 2-110-2 neurons for the LSTM and BN-LSTM, and 1-110-2 for the Phased LSTM. The oscillation periods of the Phased LSTMs are drawn uniformly in the exponential space to give a wide variety of applicable frequencies, i.e., $\tau \sim \exp(\mathcal{U}(0, 3))$. All other parameters match between models where applicable. The default LSTM parameters are given in the Lasagne Theano implementation, and were kept for LSTM, BN-LSTM, and Phased LSTM. Appropriate gate biasing was investigated but did not resolve the discrepancies between the models.

All three networks excel under standard sampling conditions as expected, as seen in Fig. 3d (left). However, for the same number of epochs, increasing the data sampling by a factor of ten has devastating effects for both LSTM and BN-LSTM, dropping their accuracy down to near chance (Fig. 3d, middle). Presumably, if given enough training iterations, their accuracies would return to the normal baseline. However, for the oversampled condition, Phased LSTM actually *increases* in accuracy, as it receives more information about the underlying waveform. Finally, if the updates are not evenly spaced and are instead sampled at asynchronous times, even when controlled to have the same number of points as the standard sampling condition, it appears to make the problem rather challenging for traditional state-of-the-art models (Fig. 3d, right). However, the Phased LSTM has no difficulty with the asynchronously sampled data, because the time gates $k_t$ do not need regular updates and can be correctly sampled at any continuous time within the period.

We extend the previous task by training the same RNN architectures on signals composed of two sine waves. The goal is to distinguish signals composed of sine waves with periods $T_1 \sim \mathcal{U}(5, 6)$ and $T_2 \sim \mathcal{U}(13, 15)$, each with independent phase, from signals composed of sine waves with periods $T_1 \sim \{\mathcal{U}(1, 5) \cup \mathcal{U}(6, 100)\}$ and $T_2 \sim \{\mathcal{U}(1, 13) \cup \mathcal{U}(15, 100)\}$, again with independent phase. Despite being significantly more challenging, Fig. 4a demonstrates how quickly the Phased LSTM converges to the correct solution compared to the standard approaches, using exactly the same parameters. Additionally, the Phased LSTM appears to exhibit very low variance during training.

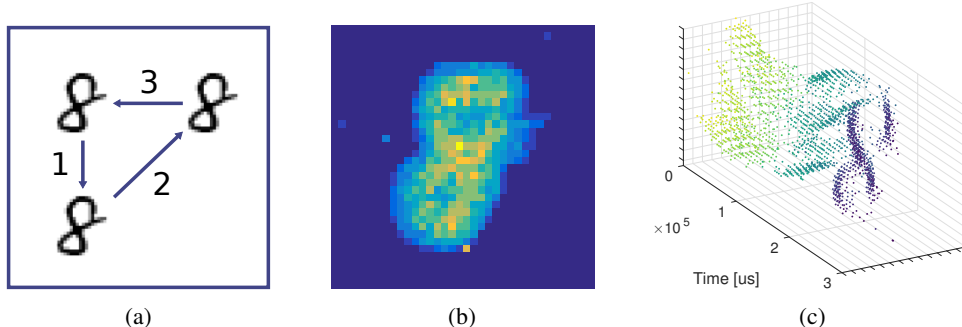

Figure 5: N-MNIST experiment. (**a**) Sketch of digit movement seen by the image sensor. (**b**) Frame-based representation of an '8' digit from the N-MNIST dataset [24] obtained by integrating all input spikes for each pixel. (**c**) Spatio-temporal representation of the digit, presented in three saccades as in (a). Note that this representation shows the digit more clearly than the blurred frame-based one.

## 3.2 Adding Task

To investigate how introducing time gates helps learning when long memory is required, we revisit an original LSTM task called the adding task [16]. In this task, a sequence of random numbers is presented along with an indicator input stream. When there is a 0 in the indicator input stream, the presented value should be ignored; a 1 indicates that the value should be added. At the end of presentation the network produces a sum of all indicated values. Unlike the previous tasks, there is no inherent periodicity in the input, and it is one of the original tasks that LSTM was designed to solve well. This would seem to work against the advantages of Phased LSTM, but using a longer period for the time gate $k_t$ could allow more effective training as a unit opens only a for a few timesteps during training.

In this task, a sequence of numbers (of length 490 to 510) was drawn from $\mathcal{U}(-0.5,\ 0.5)$. Two numbers in this stream of numbers are marked for addition: one from the first 10% of numbers (drawn with uniform probability) and one in the last half (drawn with uniform probability), producing a model of a long and noisy stream of data with only few significant points. Importantly, this should challenge the Phased LSTM model because there is no inherent periodicity and every timestep could contain the important marked points.

The same network architecture is used as before. The period $\tau$ was drawn uniformly in the exponential domain, comparing four sampling intervals $\exp(\mathcal{U}(0,2)), \exp(\mathcal{U}(2,4)), \exp(\mathcal{U}(4,6))$, and $\exp(\mathcal{U}(6,8))$. Note that despite different $\tau$ values, the total number of LSTM updates remains approximately the same, since the overall sparseness is set by $r_{on}$. However, a longer period $\tau$ provides a longer jump through the past timesteps for the gradient during backpropagation-through-time.

Moreover, we investigate whether the model can learn longer sequences more effectively when longer periods are used. By varying the period $\tau$, the results in Fig. 4b show longer $\tau$ accelerates training of the network to learn much longer sequences faster.

## 3.3 N-MNIST Event-Based Visual Recognition

To test performance on real-world asynchronously sampled data, we make use of the publicly-available N-MNIST [24] dataset for neuromorphic vision. The recordings come from an event-based vision sensor that is sensitive to local temporal contrast changes [26]. An event is generated from a pixel when its local contrast change exceeds a threshold. Every event is encoded as a 4-tuple $\langle x,\ y,\ p,\ t \rangle$ with position $x$, $y$ of the pixel, a polarity bit $p$ (indicating a contrast increase or decrease), and a timestamp $t$ indicating the time when the event is generated. The recordings consist of events generated by the vision sensor while the sensor undergoes three saccadic movements facing a static digit from the MNIST dataset (Fig. 5a). An example of the event responses can be seen in Fig. 5c.

In previous work using event-based input data [21, 23], the timing information was sometimes removed and instead a frame-based representation was generated by computing the pixel-wise event-rate over some time period (as shown in Fig. 5(b)). Note that the spatio-temporal surface of

Table 1: Accuracy on N-MNIST

|  | CNN | BN-LSTM | Phased LSTM ($\tau = 100$ms) |
|---|---|---|---|
| Accuracy at Epoch 1 | 73.81% $\pm$ 3.5 | 40.87% $\pm$ 13.3 | **90.32% $\pm$ 2.3** |
| Train/test $\rho = 0.75$ | 95.02% $\pm$ 0.3 | 96.93% $\pm$ 0.12 | **97.28% $\pm$ 0.1** |
| Test with $\rho = 0.4$ | 90.67% $\pm$ 0.3 | 94.79% $\pm$ 0.03 | **95.11% $\pm$ 0.2** |
| Test with $\rho = 1.0$ | 94.99% $\pm$ 0.3 | 96.55% $\pm$ 0.63 | **97.27% $\pm$ 0.1** |
| LSTM Updates | – | 3153 per neuron | **159 $\pm$ 2.8 per neuron** |

events in Fig. 5(c) reveals details of the digit much more clearly than in the blurred frame-based representation.The Phased LSTM allows us to operate directly on such spatio-temporal event streams.

Table 1 summarizes classification results for three different network types: a CNN trained on frame-based representations of N-MNIST digits and two RNNs, a BN-LSTM and a Phased LSTM, trained directly on the event streams. Regular LSTM is not shown, as it was found to perform worse. The CNN was comprised of three alternating layers of 8 kernels of 5x5 convolution with a leaky ReLU nonlinearity and 2x2 max-pooling, which were then fully-connected to 256 neurons, and finally fully-connected to the 10 output classes. The event pixel address was used to produce a 40-dimensional embedding via a learned embedding matrix [9], and combined with the polarity to produce the input. Therefore, the network architecture was 41-110-10 for the Phased LSTM and 42-110-10 for the BN-LSTM, with the time given as an extra input dimension to the BN-LSTM.

Table 1 shows that Phased LSTM trains faster than alternative models and achieves much higher accuracy with a lower variance even within the first epoch of training. We further define a factor, $\rho$, which represents the probability that an event is included, i.e. $\rho = 1.0$ means all events are included. The RNN models are trained with $\rho = 0.75$, and again the Phased LSTM achieves slightly higher performance than the BN-LSTM model. When testing with $\rho = 0.4$ (fewer events) and $\rho = 1.0$ (more events) without retraining, both RNN models perform well and greatly outperform the CNN. This is because the accumulated statistics of the frame-based input to the CNN change drastically when the overall spike rates are altered. The Phased LSTM RNNs seem to have learned a stable spatio-temporal surface on the input and are only slightly altered by sampling it more or less frequently.

Finally, as each neuron of the Phased LSTM only updates about 5% of the time, on average, 159 updates are needed in comparison to the 3153 updates needed per neuron of the BN-LSTM, leading to an approximate twenty-fold reduction in run time compute cost. It is also worth noting that these results form a new state-of-the-art accuracy for this dataset [24, 7].

### 3.4 Visual-Auditory Sensor Fusion for Lip Reading

Finally, we demonstrate the use of Phased LSTM on a task involving sensors with different sampling rates. Few RNN models ever attempt to merge sensors of different input frequencies, although the sampling rates can vary substantially. For this task, we use the GRID dataset [8]. This corpus contains video and audio of 30 speakers each uttering 1000 sentences composed of a fixed grammar and a constrained vocabulary of 51 words. The data was randomly divided into a 90%/10% train-test set. An OpenCV [17] implementation of a face detector was used on the video stream to extract the face which was then resized to grayscale 48x48 pixels. The goal here is to obtain a model that can use audio alone, video alone, or both inputs to robustly classify the sentence. However, since the audio alone is sufficient to achieve greater than 99% accuracy, sensor modalities were randomly masked to zero during training to encourage robustness towards sensory noise and loss.

The network architecture first separately processes video and audio data before merging them in two RNN layers that receive both modalities. The video stream uses three alternating layers of 16 kernels of 5x5 convolution and 2x2 subsampling to reduce the input of 1x48x48 to 16x2x2, which is then used as the input to 110 recurrent units. The audio stream connects the 39-dimensional MFCCs (13 MFCCs with first and second derivatives) to 150 recurrent units. Both streams converge into the *Merged-1* layer with 250 recurrent units, and is connected to a second hidden layer with 250 recurrent units named *Merged-2*. The output of the Merged-2 layer is fully-connected to 51 output nodes, which represent the vocabulary of GRID. For the Phased LSTM network, all recurrent units are Phased LSTM units.

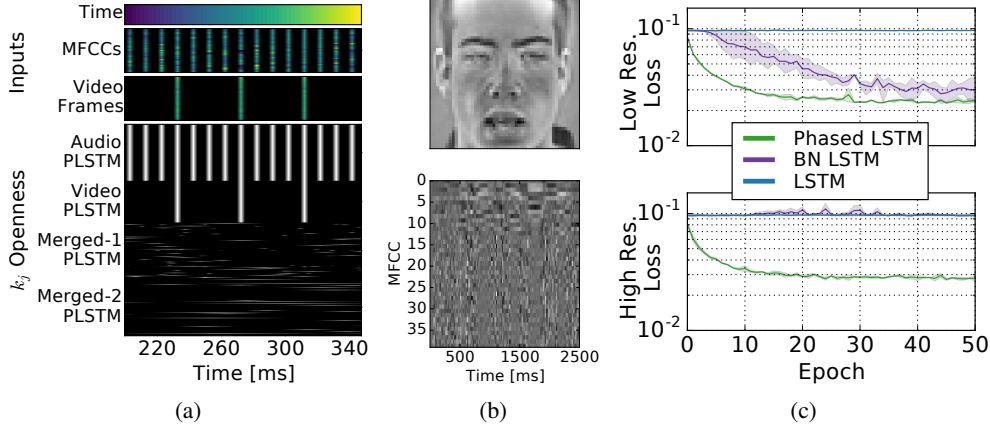

Figure 6: Lip reading experiment. **(a)** Inputs and openness of time gates for the lip reading experiment. Note that the 25fps video frame rate is a multiple of the audio input frequency (100 Hz). Phased LSTM timing parameters are configured to align to the sampling time of their inputs. **(b)** Example input of video (top) and audio (bottom). **(c)** Test loss using the video stream alone. Video frame rate is 40ms. Top: low resolution condition, MFCCs computed every 40ms with a network update every 40 ms; Bottom: high resolution condition, MFCCs every 10 ms with a network update every 10 ms.

In the audio and video Phased LSTM layers, we manually align the open periods of the time gates to the sampling times of the inputs and disable learning of the $\tau$ and $s$ parameters (see Fig. 6a). This prevents presenting zeros or artificial interpolations to the network when data is not present. In the merged layers, however, the parameters of the time gate are learned, with the period $\tau$ of the first merged layer drawn from $\mathcal{U}(10, 1000)$ and the second from $\mathcal{U}(500, 3000)$. Fig. 6b shows a visualization of one frame of video and the complete duration of an audio sample.

During evaluation, all networks achieve greater than 98% accuracy on audio-only and combined audio-video inputs. However, video-only evaluation with an audio-video capable network proved the most challenging, so the results in Fig. 6c focus on these results (though result rankings are representative of all conditions). Two differently-sampled versions of the data were used: In the first "low resolution" version (Fig. 6c, top), the sampling rate of the MFCCs was matched to the sampling rate of the 25 fps video. In the second "high-resolution" condition, the sampling rate was set to the more common value of 100 Hz sampling frequency (Fig. 6c, bottom and shown in Fig. 6a). The higher audio sampling rate did not increase accuracy, but allows for a faster latency (10ms instead of 40ms). The Phased LSTM again converges substantially faster than both LSTM and batch-normalized LSTM. The peak accuracy of 81.15% compares favorably against lipreading-focused state-of-the-art approaches [28] while avoiding manually-crafted features.

## 4  Discussion

The Phased LSTM has many surprising advantages. With its rhythmic periodicity, it acts like a learnable, gated Fourier transform on its input, permitting very fine timing discrimination. Alternatively, the rhythmic periodicity can be viewed as a kind of persistent dropout that preserves state [27], enhancing model diversity. The rhythmic inactivation can even be viewed as a shortcut to the past for gradient backpropagation, accelerating training. The presented results support these interpretations, demonstrating the ability to discriminate rhythmic signals and to learn long memory traces. Importantly, in all experiments, Phased LSTM converges more quickly and theoretically requires only 5% of the computes at runtime, while often improving in accuracy compared to standard LSTM. The presented methods can also easily be extended to GRUs [6], and it is likely that even simpler models, such as ones that use a square-wave-like oscillation, will perform well, thereby making even more efficient and encouraging alternative Phased LSTM formulations. An inspiration for using oscillations in recurrent networks comes from computational neuroscience [3], where rhythms have been shown to play important roles for synchronization and plasticity [22]. Phased LSTMs were not designed as biologically plausible models, but may help explain some of the advantages and robustness of learning in large spiking recurrent networks.

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
