[Reviews · NeurIPS 2016]

Reviewer 1

Summary

The paper enhances LSTM with a "time gate". The time gate is controlled only by time and by learned parameters. Each LSTM cell has a different time gate. When the time gate is closed, the cell is not updated.

Qualitative Assessment

I like that the paper tries something new. Technical quality: 1) The proposed formulation of the time gate is not necessary optimal. It may be better to make the time gate a continuous function of the time. E.g., the continuous time gate can be a piece-wise linear function with values: alpha, 1, alpha, 0, alpha, 1, ... 2) It would be nice to run the model on a speech recognition benchmark. Novelty: It is nice that the paper considers processing of asynchronous events. Impact: The work is inspiring. People can propose improvements. The results show significant improvements on the tried tasks. Clarity: It is not clear whether the twenty-fold reduction in run time compute cost is just theoretical or measured on a hardware.

Confidence in this Review

2-Confident (read it all; understood it all reasonably well)


Reviewer 2

Summary

This paper introduces the Phased LSTM, a structure which is meant to handle irregularly sampled data through the introduction of a new time gate. The authors test PLSTMs on 4 datasets, showing improvement over standard LSTMs. In the case of the N-MINST and Lip Reading datasets, an improvement is achieved over the state-of-the-art.

Qualitative Assessment

Overall, this seems like a great paper. There is a sufficient degree of novelty in the new cell structure and its usefulness has been proven. Given the prevalence of time series data and to learn representations even when this data is not cleanly sampled, I believe this technique could have high impact. The paper is also nicely written and the presentation is clear. Minor questions/comments: How many samples were used to learn the models in the frequency discrimination task and how does accuracy change with added data? Could Phased LSTMs be stacked to further improve performance? In Figure 4, please use the same color for Phased LSTM in the two plots. It would be useful to have an image of the deep learning structures used in the experiments. If this doesn’t fit in the main paper, an appendix or extended version should be provided.

Confidence in this Review

2-Confident (read it all; understood it all reasonably well)


Reviewer 3

Summary

LSTMs and related models (e.g GRUs) have become the workhorses of neural network research and applications involving sequences. The key idea of LSTMs is to use gating over the memory cells to mitigate the "vanishing gradients" problem that occurs in standard recurrent networks, allowing the network to learn tasks that require persisting memories over many timesteps. This work proposes an extension to LSTMs, adding "time gates" which control updates to the memory. These gates are controlled by oscillators, which only allow updates to the memory during certain points in the oscillation. The periodicity and phase-shift of the gates are learned during training. One additional property these time gates have, is that as they are controlled by a continuous-time oscillator, the value of the gates can be sampled at irregular time intervals. These phased LSTMs are tested in 4 experiments: classifying sinusoidal signals (a task tailored to the properties of this model), adding task (similar to the adding task in the original LSTM paper), event-based encoding of MNIST and a lip-reading dataset. In all of these tasks the phased LSTMs outperform the original LSTM.

Qualitative Assessment

LSTMs (and GRUs) are increasingly used as basic building blocks in neural network architectures, both in inherently sequential problems but also in other applications as many other problems can usefully be decomposed into sequential problems using mechanisms such as attention. Despite being devised some time ago, LSTMs have proved to be difficult to beat as a general purpose tools for modeling sequential structures (e.g. [1]). This paper presents an interesting idea for improving the performance of LSTMs, particularly on tasks which contain cyclical structure. It is novel and explains the model and motivations well. There are aspect of the analysis and experimental results which could be improved on, but it is a novel approach that will be of interest to the field. I have several suggestions for improvements below, but these do not significantly detract from the work, which is of a high standard. One aspect that should be clarified is exactly how time-sampling was performed in the cases where sampling was "asynchronous." It's clear that k_t can be computed for any time t, but the remainder of the LSTM still appears to be a discrete-time update (for example, two updates in quick succession when k_t ~ 1 will affect c_t differently than only 1 update is sampled during the k_t~1 phase). It would be very interesting if the authors could include an any analysis or insight into the solution their model is learning on simple tasks such as adding or classifying sinusoids. The manuscript mentions the potential for reduced computational load, particularly for high-frequency signals, due to the gating alleviating the need to compute update to the cell when k_t = 0. It would be ideal if this was supported by empirical analysis in one of the experiments at test time (where alpha = 0). Although realized in a different way, these ideas bear some resemblance to reservoir computing. In particular, both maintain memories using fixed oscillators while learning the connections into/out of the oscillators. It would be helpful to cite some of this work and mention this connection/distinctions. The choice of experimental comparisons is somewhat eclectic. It certainly makes sense to compare with one of the original LSTM tasks (adding). It would help to assess the potentially impact of this work if it were used on a larger, standard sequential benchmark that doesn't necessarily contain obvious oscillatory structure. For example, a language modeling or machine translation task. would improve the model substantially to compare with a more commonly used sequential benchmark (e.g. [3]). One aspect of the model which the authors mention is the tendency to preserve memories over many timesteps due the gating effect of the time gates (particularly if tau is large). One might hope to obtain something similar in a standard LSTM by heavily biasing some of the forget gates to 1 and the input gates to 0. It would be a useful comparison to check that simple initialization tricks can't reproduce the results. [1] LSTM: A Search Space Odyssey http://arxiv.org/pdf/1503.04069v1.pdf [2] A clockwork RNN. http://arxiv.org/abs/1402.3511 [3] Recurrent neural network regularization http://arxiv.org/pdf/1409.2329v5.pdf ** After feedback ** Due to the large number of reviews, there wasn't space to address my comments in depth. Overall, my ratings are unchanged (they were already high), but I agree with the bulk of the other reviewers this is a good paper.

Confidence in this Review

2-Confident (read it all; understood it all reasonably well)


Reviewer 4

Summary

This paper proposed a variation of LSTM model by introducing a time moderator gate, so that the proposed model can handle longer sequence input and keep memories for longer time. The newly designed time gate is a cyclic function of time and controls the degree to which the cells and hidden states are updated based on the current input or kept from the previous time step. Empirical results demonstrated the superiority of the new model compared with standard LSTM by capturing longer time memories.

Qualitative Assessment

Improving RNN’s capacity of capturing longer memories and thus handling longer sequences is a quite important problem. This paper proposed an interesting and effective idea on new gates whose value is computed by the input time. With different each cell in LSTM has the ability of capturing memories in different lengths, as illustrated in figure 2(b), thus it is suitable for longer sequence of over-sampled sequence input. Also the same idea can be generalized to other gated RNN such as GRU. In the meanwhile, I have a few concerns as follows: I am agreed that the proposed model can handle long sequences, but I’m not convinced that the designed time gate explicitly handles the event-based or irregularly sampled data. The purpose of introducing a time gate, from my perspective, is mainly about to limit the hidden states to be updated only at a small ratio of time. To be convinced that the proposed time gate is good for irregular or event-based sequence, I’d like to see the comparison of the proposed one with an almost same but random time gate: That is, the time gate k_t is just a random number with the same distribution as it is (e.g., with prob. r_on to be 1 and other to be 0, or other ‘leaky’ modifications) but independent of input time t. This random model is probably able to handle long sequence as well since the open ratio is also small. I’d like to see whether the propose model can model event-based sequences better. At this time point, the input time to the gate k_t is more like just a random seed to generate the gate value with mean (r_on) and does nothing related to the time step itself. In experiments, Figure 3(d) showed the proposed LSTM outperformed baselines in both _high resolution sampling_ and _async. sampling_ settings. But what is the average sampling rate in the async setting? It seems that even async setting has more input steps than standard condition. Then how can we tell if the baselines fail in this condition just because of the longer sequences than standard condition? The authors claimed that the new LSTM has ‘an order-of-magnitude fewer computes’. I’m not sure the computes that they referred to are the numbers of training epochs before convergence, or the computing time for applying the model on one input sequence. From the descriptions in introduction section, the authors aim to handle _an unnecessarily higher computational load and power consumption so that all units in the network can be updated with one time step_. That is, the new model is desired to have efficiency by only partially updating its cells at each time step. However, this might not be true for the proposed model. First, during training part due to the ‘leaky’ behavior, all nodes are needed to be updated. Second, even only a small portion of states need to be updated, I’m not sure this can lead to more efficiency, since commonly neural network operations is done by matrix operations (especially on GPU), and whether one element is skipped or not doesn’t make much sense. At the end of Section 3.3 the authors mentioned that the proposed LSTM needs 5% of updates per neuron, but this may not lead to 5% time. Or the authors should point out if their models are all implemented in other efficiently way.

Confidence in this Review

2-Confident (read it all; understood it all reasonably well)


Reviewer 5

Summary

This paper proposes a novel LSTM modulating architecture called "Phased" LSTMs which adds multiple period masks to LSTM networks with different opening frequencies therefore enabling a faster learning of long term dependencies. This strategy, somewhat akin to a Fourier decomposition, presents many advantages besides providing a faster training of LSTM networks with state of the art performance on various tasks. It stand outs by its ability to enable LSTM to work with asynchronous data feeds. This means that the inputs need not arrive with a regular sampling to the network. The authors carefully explain the motivations and design of the architecture. They later on proceed with three very different experiments on different tasks which highlight that Phased LSTMs are able to feed on asynchronous data, offer a better than standard LSTMs in this setting, and train faster in all experiments.

Qualitative Assessment

This paper is nice to read, well presented, and presents a compelling innovation in the field of recurrent neural nets which has deep ties with the theory of frequency decomposition of time series data. The impact of phased LSTMs, because of their ability to handle asynchronous data feeds, is potentially huge. Most modern sensing problems deal with asynchronous time series data feeds. The phased input gate is an elegant idea, somewhat similar to considering the projection of a time series on a Fourier basis, which clearly improve the speed at which LSTMs can learn by considerably reducing the number of updates necessary and therefore allowing full propagation in time of gradients. The experiments show significant improvement over pre-existing methods and seem to be conducted in a sound and methodical manner on compelling and varied tasks. The writing quality is excellent. I just have two quick remarks: Equation (11) and the corresponding section are nice simplifications that explain why the Phased LSTM may be more likely to capture long range dependencies. However, the model, as it is simplified, is similar to an AR model which is in my opinion a somewhat extreme simplification of a LSTM model. I think the authors should state that this section is over-simplified to serve a pedagogical purpose. In the experiments, I could not find how many different oscillation periods were used in practice. How many are sampled from the exponential distributions? This paper was nice to read, straightforward yet innovative. It is, in my opinion, worthwhile presenting to the community for its impact on neuromorphic computing and time series analysis.

Confidence in this Review

3-Expert (read the paper in detail, know the area, quite certain of my opinion)


Reviewer 6

Summary

The authors propose an extension of LTSM which allows for continuous time data to be processed by the LTSM which otherwise---as other RNNs---is restricted to discrete time data. The approach is succesful and gives good results for several data sets.

Qualitative Assessment

The paper is clear and well written. The results are impressive * I do not think that the computational complexity will increase very much but this is not clear from the paper. Perhaps a comment on the time consumed, for instance reported as epochs/s? * The accuracy is very high even for the first Epoch. Is there any prior information going into the training? * Could you provide better intuition for what the gates do and why they work? * Are the nodes with different time-gates independent? If so, are you not training separate networks? Why is this then different from using several normal networks with the input data lagged?

Confidence in this Review

2-Confident (read it all; understood it all reasonably well)